# Natural resources modulate the nexus between environmental shocks and human mobility

Michael Brottrager[1], Jesus Crespo Cuaresma [2,3,4,5] ✉, Dominic Kniveton [6,7] & Saleem H. Ali [7,8]

In the context of natural resource degradation, migration can act as means of adaptation both for those leaving and those supported by remittances. Migration can also result from an inability to adapt in-situ, with people forced to move, sometimes to situations of worse or of the same exposure to environmental threats. The deleterious impacts of resource degradation have been proposed in some situations to limit the ability to move. In this contribution, we use remote sensed information coupled with population density data for continental Africa to assess quantitatively the prevalence of migration and immobility in the context of one cause of resource degradation: drought. We find that the effect of drought on mobility is amplified with the frequency at which droughts are experienced and that higher income households appear more resilient to climatic shocks and are less likely to resort to mobility as an adaptation response.

Over the last 30 years, there has been a rising tide of predictions of mass human migrations, either forced or by choice, in the face of climate change[1]. While it is accepted that many people are currently displaced by climate-related disasters[2], there is also a credible suggestion that migration might become less rather than more prevalent with future climate change[3]. The logic behind this latter possibility is that the impacts of climate change are likely to reduce the assets of vulnerable populations, impeding their ability to move[4].

From a theoretical perspective, the role played by natural resources as a mediator between environmental shocks and migration decisions is complex[5]. The so-called "environmental scarcity" hypothesis poses that risks associated with environmental shocks and their variability make migration more likely, with households reallocating to compensate for potential losses in natural capital. The "environmental capital" hypothesis, on the other hand, sees natural capital as a resource providing income that in turn may support (long-distance) migration decisions. To the extent that population immobility in the

context of environmental shocks and climate risks can be explained by a lack of resources, access to income streams from natural capital appears important to explain migration patterns and thus the potential emergence of trapped populations[6]. Natural resources play a mediating role in both determining who is vulnerable to climate change and who is able to afford to migrate away. The evidence of the extent of the phenomenon of immobility is restricted to selected regions or nations[7–11]. Some studies dealing with immobile populations exist[12,13]. These studies explore the sensitivity of international migration, measured through bilateral flow data, to temperature increases in origin countries. The analysis reveals that, while for middle-income economies increased temperatures are associated with increased migration rates to other countries, in poorer countries the reverse is observed[13]. The literature finds that, in response to temperature shocks, international migration flows tend to decrease for roughly 5 years before they increase for more than 20 years. In this study, we use the relative changes in measures of subnational migration in response to

[1]Department of Economics, Johannes Kepler University, Linz, Austria. [2]Department of Economics, Vienna University of Economics and Business, Vienna, Austria. [3]Population and Just Societies Program, International Institute for Applied Systems Analysis, Laxenburg, Austria. [4]Wittgenstein Centre for Demography and Global Human Capital, Vienna, Austria. [5]Austrian Institute of Economic Research, Vienna, Austria. [6]School of Global Studies, University of Sussex, Brighton, UK. [7]United Nations International Resource Panel, Paris, France. [8]Department of Geography and Spatial Sciences, University of Delaware, Newark, NJ, USA. ✉e-mail: jcrespo@wu.ac.at

environmental stress (drought) to identify immobile populations and quantify the effect of environmental shocks on internal migration.

A schematic contextual framework to understand how resource scarcity and abundance intersect with migration processes is presented in Fig. 1. This diagram shows that resource scarcity and resource abundance can both lead to disruptive impacts on human populations and their well-being. Different causal pathways are presented in this regard and conflict can be a result of these divergent mechanisms. Demography links to environmental factors through metrics such as carrying capacity (which can itself change with adaptive technologies) and may lead to resource scarcity. On the other hand, resource abundance without appropriate distributive mechanisms can lead to inequality and hamper economic development, which is in turn also linked to migration movements. Conflict can occur within the migration nexus through the combination of tribalist impulses ensuing from resource scarcity or distribution failures, particularly in kleptocracies, as well as through the kinetics of physical movement and clashes with existing populations. Containing these scenarios to avert conflict requires us to overcome the challenges of governing an inherently complex system and also the security imperatives which any conflict dynamic can instigate. Two overarching challenges frame this set of relationships: political governance at the macro level and security and management concerns at the micro level.

In our contribution, we examine the agricultural pathway of environmental risks to migration, highlighting the mechanism of deteriorating land resources due to drought. In particular, we explore the statistical relationship between drought episodes (identified by making use of the Standardized Precipitation Evapotranspiration Index (SPEI)) and subnational population density, thus linking environmental shocks to (internal) migration flows. We explore whether the presence of mineral resources acts as a mediating factor in the relationship between drought and migration. Focusing on the continent of Africa, we find that for locations with low income, drought conditions are statistically related to increased internal migration from the affected area, while for international migration they are associated with decreased international migration (see Supplementary Material). We also find that sustained drought episodes are associated with

stronger effects, and that the presence of mineral resources dampens the influence of drought on migration.

Sub-Saharan Africa has received great attention from researchers working on the empirical linkages between climate and human mobility. Most of the existing studies have analyzed the year-on-year correlation between weather phenomena and migration. In multi-country studies of Sub-Saharan Africa, the link between average rainfall and urbanization has been analyzed[14], as well as how temperature and precipitation anomalies affect migration outcomes[15]. In this strand of literature, there is no clear empirical consensus on the relationship and the direction of association between climate and migration. Climatic shocks may induce migration on the one hand and constrain human mobility on the other. Cross-national studies based on household surveys and micro-censuses report mixed evidence: whilst an increased temperature is associated with higher international migration in Uganda, outmigration decreases with temperature rise in Burkina Faso and Kenya, and no relationship is found between migration and temperature anomalies in Nigeria and Senegal[16,17]. Looking beyond Africa, country studies do not tend to find a consistent pattern of association. For example, rainfall deficits suppress US-bound migration from rural Mexico according to some studies[18,19] but increase migration according to others[20]. Likewise, macro-level studies of bilateral migration between countries also report inconsistent findings with international migration increasing with higher temperature on the one hand[13,21] and not affecting migration on the other[22]. Meta-analytical results confirm the heterogeneity of effects reported in the empirical literature on the climate-migration link[23].

The decision to migrate is the result of complex reasoning and is influenced by external factors such as poverty, social and political exclusion, conflict, labor requirements, as well as many household characteristics (size, income, landholding, aspiration). To analyze such decisions, economists often use modeling frameworks that build upon comparisons of emigration costs and potential gains from migrating. Emigration costs might be interpreted as monetary (e.g., cost of relocating) or non-monetary (e.g., psychological) costs. Environmental degradation through a drought can act as a push factor for migration for some households (by reducing current and present agricultural

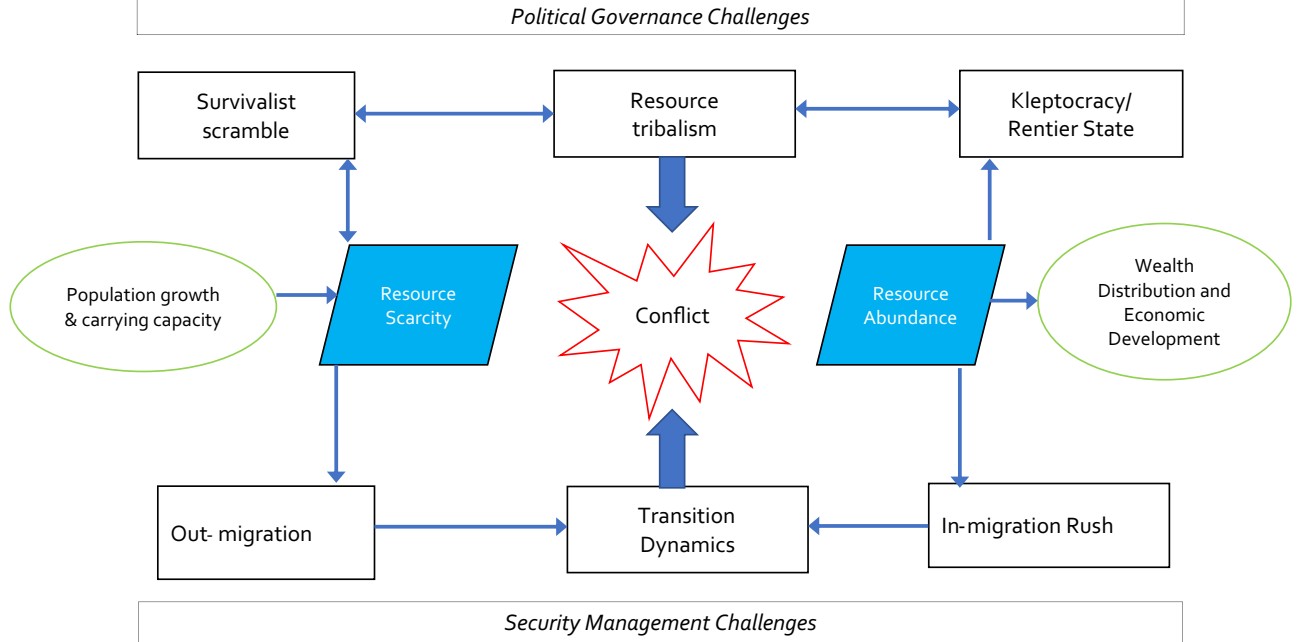

**Fig. 1 | Conceptual framework of the interaction between resource scarcity/abundance and migration.** Natural resources as a mediator in the link between environmental shocks and migration.

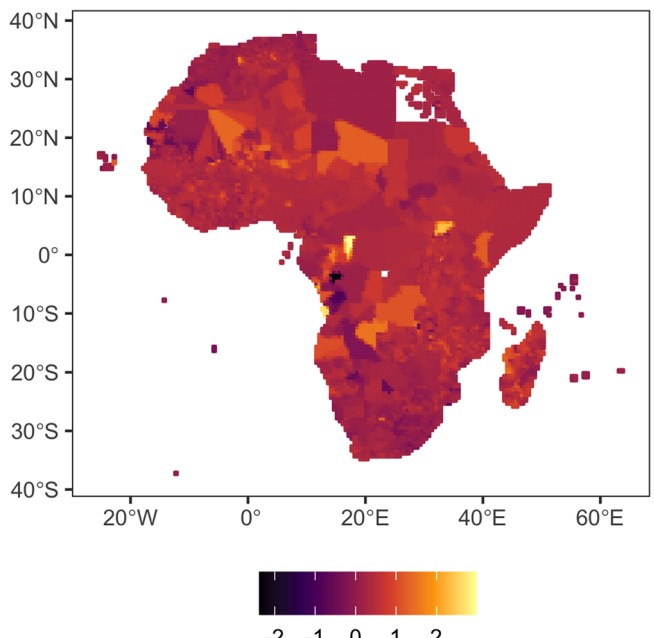

**Fig. 2 | Population growth at the grid-cell level in Africa, 2005–2015 (log differences over the full period).** Source: Gridded Population of the World version 4.

returns), but it can also hamper mobility for others by lowering the economic resources necessary for migrating. The impact of drought on migration is indirectly mediated through physical, economic and political factors which are in turn affected by environmental change[3,24,25]. In order to identify a potential relationship between drought, land degradation and migration decisions, we make use of disaggregated (grid-level) data on migration and population density. Specifically, we collect information on our variables of interest both at the national and at the $0.5 \times 0.5$ degrees cell level (approximately 55 km at the equator). Such a disaggregated approach accommodates other phenomena of human mobility, and in particular internal migration (typically not accounted for in analyses at the country level). It should be noticed that our level of granularity is not able to capture within-cell human mobility. We are therefore unable to perfectly explore internal migration to urban areas from close rural neighboring regions, or across population agglomerations that are contained in single cells.

## Results

The main source of data for the grid-level analysis is the Gridded Population of the World (GPW) collection, now in its fourth version (GPWv4). This dataset models the distribution of the human population (counts and densities) on a continuous global raster surface and we match it with climate data of a similar resolution, thus effectively dropping locations that are not inhabited, or are projected to be inhabited. Since the release of the first version of this global population surface in 1995, the essential inputs to GPW have been population census tables and corresponding geographic boundaries. The purpose of GPW is to provide a spatially disaggregated population layer that is compatible with datasets from social, economic, and earth science disciplines, as well as remote sensing. GPWv4 is a raster data collection of globally integrated national population data from the 2010 round of Population and Housing Censuses, which occurred between 2005 and 2014. The input data are extrapolated to produce population estimates for the years 2000, 2005, 2010, 2015, and 2020. One major drawback of using these data to infer migration trends is that the population changes estimated in the GPWv4 data are also affected by changes in birth and death rates. In our main specification, we assume these rates

to remain stable within cells (around a global trend) over the observation period and thus be captured by the cell fixed effects. This implies that immigration effects may be overestimated in locations where birth rates increase or mortality fall in a given period with respect to the global trend observed, with the opposite holding for emigration effects. The nature of the data used also does not allow us to assess different types of migration (long-term vs. short-term mobility, or return migration patterns). Given the fact that census information is the main source of GWPv4, the dataset also presents limitations concerning the potential exclusion of persons in some groups (refugees, internally displaced people or nomadic populations) that may, in turn, be particularly vulnerable to climatic shocks and affected by immobility[26]. The change in population at the grid-cell level used in our analysis for the period 2005–2015 is depicted in Fig. 2. Differences in population changes are large both across and within nations of the continent, and low levels of population change can be found in many areas. The statistical analysis carried out aims at understanding the heterogeneity of human mobility reactions to environmental shocks as mediated by income and natural (mineral) resource availability.

The effect estimates of the response of international migration at the country level to variations in growing season SPEI trajectories based on established econometric models[13] are presented in the Supplementary Material. For our analysis of international migration, we use data on migrant stocks spanning the period from 1960 to 2000[27]. Those stocks are converted into migration flows by summing all net flows for the same countries of origin and computing emigration rates as the ratio between the aggregate net flow of emigrants in the decade relative to the origin country population at the beginning of the decade. The main advantage of these data is that their main sources are national censuses, which are much more accurate in counting foreign-born individuals as compared to flow measures. As the data are only available every 10 years, migration responses capture long-term trends. For our international analysis, we compare SPEI trajectories across countries and decades. We exploit differences in period-average SPEI scores compared to long-term levels of SPEI in periods that range from a length of 2–10 years. Comparing effect size estimates across those period comparisons allows us to shed light on long-run migration responses to worsening climate conditions and by extension a proxy of worsening land-based natural resources. Our results show statistically significant effects of drought on migration decisions[13]. The reaction of emigration rates, however, is exclusively associated with relatively poor countries and particularly sizable for long-lasting droughts. A potential channel explaining this effect is related to worsening economic conditions creating obstacles to outmigration[28]. In economies in which agricultural productivity is so low as to leave rural populations liquidity constrained and constrained to the primary sector, a worsening (improving) climate and lower (higher) agricultural productivity may actually slow (increase) economic transformation and economic growth, thus contributing to poverty traps.

In order to investigate the potential tempering effects that resource availability as an alternative source of income might have on negative climate shocks, grid-level data sourced from recent surveys are employed in our analysis.

Table 1 displays the estimates based on the specification relating drought and land degradation to log-transformed grid-level population density, using data spanning the period 2000–2015 in 5-year intervals. Column 1 displays a naive correlation of our drought measure with log-population levels without controlling for any time or cell fixed effects. By not including cell and time fixed effects, we implicitly allow for cell and time-specific confounding factors to bias our estimation results. For example, if cells in rural areas are more likely to experience both drought and strong migration toward urban areas, we would have to add cell fixed effects as this higher migration tendency would create a spurious correlation in models that do not account for

**Table 1 | Effects of drought events (columns 1–4) and soil degradation (column 5) on population**

|  | (1) Drought | (2) Drought | (3) Drought | (4) Drought | (5) Soil |
|---|---|---|---|---|---|
| Event | −6.649*** | −0.549*** | −0.127*** | −0.126*** | −0.151 |
|  | (1.050) | (0.192) | (0.010) | (0.039) | (0.224) |
| Cell FE | No | Yes | Yes | Yes | Yes |
| Year FE | No | No | Yes | Yes | Yes |
| HAC SE | No | No | No | Yes | Yes |
| Obs | 41,868 | 41,868 | 42,064 | 41,868 | 41,868 |
| $R^2$ | 0.095 | 0.0419 | 0.0046 | 0.004 | 0.004 |

Standard errors in parentheses. Within-$R^2$ if cell fixed effects included.

***$p < 0.01$, based on two-sided $t$-tests.

such time-invariant unobservables. Confounding factors that are associated with time would be period-specific events—such as El Nino or region-wide conflict potentially spurring migration—that affect numerous cells at the same time.

The estimation results presented in columns 1–3 give evidence of the importance to control for space- and time-invariant confounders, which is accounted for in the regression model by including cell fixed effects in column 2 and cell as well as time fixed effects in column 3. Part of the effect of droughts on migration might be due to spatial correlation, since drought phenomena are hardly cell-specific but also affect neighboring cells. To account for spatial correlation, the results presented in columns 4 and 5 utilize standard errors which are estimated with a spatial heteroskedasticity and autocorrelation consistent (HAC) correction, allowing for both cross-sectional spatial correlation and location-specific serial correlation through the method developed by ref. [29]. The negative effect of drought on population is reduced in absolute value as controls are included, and accounting for spatially correlated shocks decreases the uncertainty surrounding the estimated effect.

Column 4 displays our preferred specification, including cell and time fixed effects, as well as HAC standard errors, which shows a statistically significant and sizable negative effect of prolonged drought periods on cell-level population. As our main explanatory variable is measured in fractions of a year, dividing the parameter estimate by 12 gives us the effect of 1 additional month of drought, which would decrease the population in that cell by about 1%. Given the mean

population by cell is about 91,966, a reduction of 1% accounts for about 919 persons per cell, and a total reduction of 9.8 million people due to migration across all grid cells assuming constant birth and death rates. Column 5 presents the estimation results of a specification that relates soil carbon degradation to potential migration responses. Notwithstanding the difficulty of obtaining reliable estimates of soil carbon changes (see https://trends.earth/docs/en/background/understanding_indicators15.html), the variable does not appear significant in our regression model.

To explore potential non-monotonic effects of drought severity, the left panel of Fig. 3 displays the effects of different levels of drought duration on migration measures. Drought levels rank from 2 to more than 6 consecutive months with SPEI levels below −1.5, which are then compared to cells experiencing less than 2 months of drought. More intense drought periods tend to have more severe (average) impacts on migration, lending support to existing evidence that long-lasting declines in rainfall tend to increase migration[30]. However, the rather low precision of our drought severity estimates does not allow for any inference of non-linear changes in absolute effect sizes. It should be noticed that the method does not allow us to differentiate between recurring, seasonal multi-year droughts and single drought periods at irregular frequencies. The potential differential effect of these two types of events, which differ in terms of predictability and thus on the possibility of enacting adaptation strategies, can therefore not be teased out from our empirical analysis.

We proceed by analyzing potential heterogenous resilience to drought periods depending on income by interacting our main explanatory variable with estimates of purchasing power parity adjusted gross cell product evaluated in the initial year of our observational period. These projections are obtained from the latest version of the Global Gridded Geographically Based Economic Data[31]. The data are computed by spatial rescaling based on existing figures from subnational administrative units using a proportional allocation rule based on cell population and area. The resulting effect estimates by income quartile are displayed in the right panel of Fig. 3. Relatively richer cells are far more resilient towards drought periods compared to the poorest cells in the income distribution, lending support to some existing empirical results on the effect of negative economic shocks on internal migration patterns in poor households[32]. Part of the heterogeneity might be explained by the fact that agricultural dependence is most prevalent in poor cells and alternative sources of income tend to be more accessible in higher-income cells.

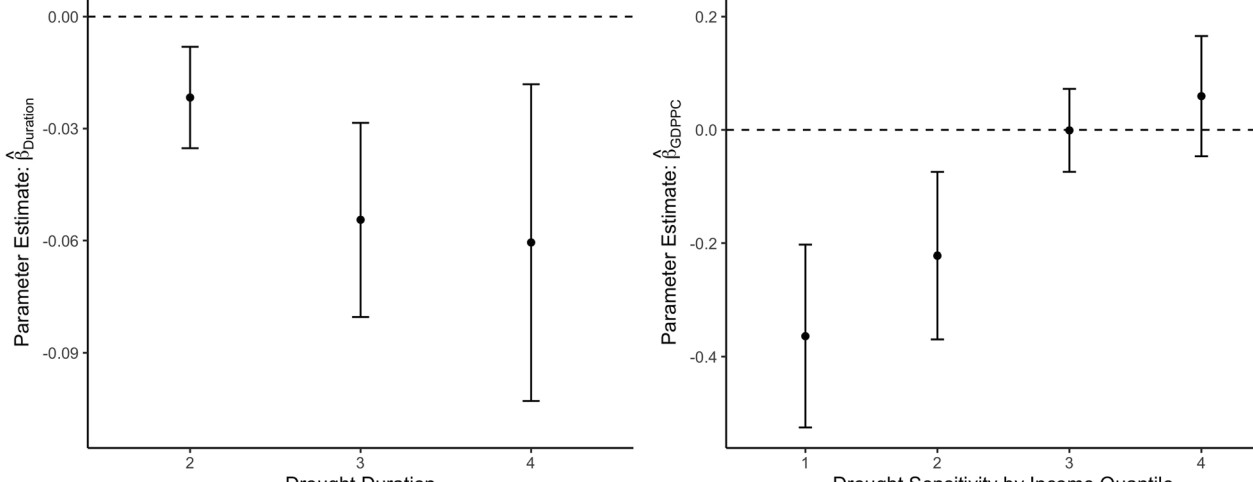

**Fig. 3 | Non-monotonic effects of drought by severity (left) and income quantiles (right).** The parameter estimates shown correspond to the mean effect ± twice its standard deviation for the regression model including cell fixed effects, time fixed effects and HAC standard errors. $n = 41,868$.

**Table 2 | Effects of drought events on population: distance to the capital city and urbanization as mediators**

|  | (1) | (2) |
|---|---|---|
| Drought event | −0.117** | −0.143*** |
|  | (0.052) | (0.038) |
| × Closeness | −0.008 |  |
|  | (0.013) |  |
| × Urbanization |  |  |
|  |  | 0.043** |
|  |  | (0.017) |
| Cell FE | Yes | Yes |
| Year FE | Yes | Yes |
| Obs. | 42,664 | 42,672 |
| $R^2$ | 0.006 | 0.006 |

HAC standard errors in parentheses.
**$p < 0.05$, ***$p < 0.01$, based on two-sided $t$-tests.

**Table 3 | Effects of drought events on population: mineral resources as mediators**

|  | (1) Mineral resources | (2) MRDS site |
|---|---|---|
| Drought event | −0.145*** | −0.151*** |
|  | (0.039) | (0.039) |
| × Minerals in cell | 0.157* |  |
|  | (0.084) |  |
| × MRDS extraction site |  | 0.283*** |
|  |  | (0.097) |
| Cell FE | Yes | Yes |
| Year FE | Yes | Yes |
| Observations | 42,672 | 42,672 |
| $R^2$ | 0.006 | 0.007 |

HAC standard errors in parentheses.
*$p < 0.1$, ***$p < 0.01$, based on two-sided $t$-tests.

The relatively higher resilience of "richer" cells might also be due to their relatively higher urbanization rate. To investigate whether our interpretation of the effects of droughts being more severe for rural areas is adequate, Table 2 displays the estimation results of a specification where our baseline drought effects interacted with the distance to the capital and the urbanization rate, measured as a percentage share of the cell considered to be urban. Table 2 shows that relatively more urbanized cells tend to be more resilient to droughts as compared to cells with rural attributes. This provides evidence that the lack of alternative means of income in rural regions appears to be one of the main drivers of the negative drought effect on migration.

Another potential alternative source of income is related to the presence of mineral resources. Developing countries have access to some of the world's largest oil and mineral reserves. They are among the largest producers of key minerals and account for most of the recent growth in mineral production[33]. The existing empirical literature suggests that an abundance of natural resources may fail to improve living standards, or even hinder economic performance, especially in the presence of weak institutions[34]. Most of the evidence, however, comes from aggregate data at the country level and offers little guidance about the local economic effects of resource abundance. In our setting, mineral resources might actually provide income in times when agricultural yields dwindle in the face of a drought shock. In line with ref. [35], we find positive effects of mineral resource presence within a cell on the sensitivity to SPEI changes (see Table 3).

**Table 4 | Effects of drought events (columns 1–3) and soil degradation (column 4) on population: specifications including country-year effects**

|  | (1) Drought | (2) Drought | (3) Drought | (4) Soil |
|---|---|---|---|---|
| Event | −0.127*** | −0.126*** | 0.003 | 0.324 |
|  | (0.010) | (0.039) | (0.029) | (0.208) |
| Cell FE | Yes | Yes | Yes | Yes |
| Year FE | Yes | Yes | Yes | Yes |
| Country × Year FE | No | No | Yes | Yes |
| HAC SE | No | Yes | Yes | Yes |
| $R^2$ | 0.004 | 0.004 | 0.004 | 0.002 |

Standard errors in parentheses.
***$p < 0.01$, based on two-sided $t$-tests.

That is, adverse drought effects are dampened by the presence of mineral resources, probably due to the possibility to gain access to an alternative source of income.

The results in Table 4 present estimates of our basic specification including country-year fixed effects that account for differences in population change across countries and over time. The population decreases in grid cells affected by drought found in specifications with cell and year fixed effects disappear once country-year fixed effects are added to the model. This indicates that drought occurrence is able to satisfactorily explain between-country variation in migration, but its effect is not statistically significant once country-wide shocks are accounted for. Given the importance of the primary sector in many of the countries of our sample, such a result may be related to the correlation between drought shocks and aggregate economic performance at the macroeconomic level, as well as to other time-varying nation-specific institutional characteristics. Studying the role played by differences in institutional settings across African countries as a determinant of the migration response to environmental shocks goes beyond the analysis carried out in this piece, but constitutes a particularly promising avenue of further research building upon the empirical results presented.

When expanding the regression specification used to the inclusion of effects that are mediated by both income and resource availability, it is the income gradient that is able to significantly explain differences in the response of mobility to drought shocks (see Supplementary Material). Figure 4 presents the implied parameter estimates for the grid cell-specific effect of drought shocks on migration based on this expanded model which accounts for both income and resource mediation. The results highlight the high degree of heterogeneity in migration responses implied by the data, with dominant emigration effects being more prominent in landlocked regions and strong differences in the effect being observed both within and between countries of the African continent.

## Discussion

In our analysis, we provide evidence that natural resources play a central role in helping human settlements cope and adapt to climate change, yet are also sensitive to the very changes they act as protection to. Our results indicate that mobile populations would be expected to show increased migration both sub-nationally and internationally in times of environmental stress. In policy terms, immobile populations, and in particular trapped populations (those where the need and aspiration to migrate are not met by the capacity to do so) can be considered as facing the greatest risks of climate and environmental change and hence needing particular support[3]. International coordination of ecological and social data needs to be prioritized within early warning systems domestically in countries and at the International Organization for Migration to ensure more efficient decisions that

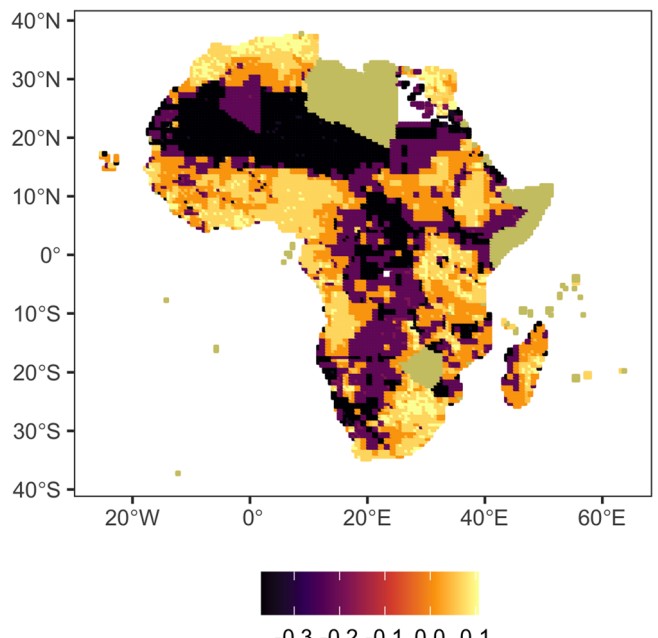

**Fig. 4 | Estimates of the effect of drought events on migration, by grid-cell.**
Missing information corresponds to countries for which data on GDP are not
available.

mitigate negative impacts on vulnerable human populations. Fur-
thermore, our results provide input for the discussion of evidence-
based policies related to natural resources governance. Participatory
processes related to natural resource stewardship and their effect on
building adaptive capacity against climate risks have often been stu-
died in the literature[36–38]. Our analysis provides a first set of results that
inform the policy discussion on such institutional changes in the
ownership structure of natural resources about the potential costs and
benefits related to mobility reactions to environmental shocks. In
particular, our methodological framework and results can help reduce
uncertainty in the estimation of the population mobility response after
environmental shocks, and provide helpful evidence for the design of
efficient models of ecosystem stewardship.

Further efforts in data collection would be necessary to inves-
tigate the effects of environmental shocks on different types of
mineral resources and migration patterns. Heterogeneity of effects
depending on the particular type of resource considered is expected
due to differences in vulnerability to climate risks and in the mag-
nitude of rents that can be obtained from their exploitation. Recent
contributions based on the use of administrative data are able to
offer a much more detailed account of differences in migration
patterns[39,40], and the combination of this information with geo-coded
environmental data may prove helpful in future research. In parti-
cular, assessing differences in the reaction to environmental shocks
in terms of permanent versus temporary human mobility appears
relevant in this context.

Similarly, a more rigorous analysis of the spatial spillovers that
environmental shocks can have appears as a promising avenue to
expand the results presented here. The estimation of statistical
models with an explicit spatial autoregressive structure to the data-
set analyzed in this contribution may shed new light on the propa-
gation of mobility phenomena across space after climatic shocks take
place. Notwithstanding the difficulty of assessing empirically spatial
linkages in the framework of the natural resource nexus, the quan-
tification of spillovers across space can be particularly important for
the design of effective regional policies to support climate change
adaptation.

## Methods

The analysis of the drivers of migration flows builds upon an estima-
tion strategy, where migration responses are regressed on a drought
indicator and potential mediators. The reduced-form regression
model linking drought and migration is given by

$$Pop_{j,t} = \alpha + \gamma_1 C_{j,t} + \gamma_2 D'_{j,t} + \gamma_p C_{j,t} \times D'_{j,t} + \phi_t + \theta_j + \varepsilon_{j,t}, \quad (1)$$

where $Pop_{j,t}$ captures the log-population measures at grid unit $j$ in time
$t$, $C_{j,t}$ captures the location-specific drought event, and $D'_{j,t}$ and
$C_{j,t} \times D'_{j,t}$ capture the mediating effects of the variable in $D'_{j,t}$ on the
effect of droughts on migration, while $\phi_t$ and $\theta_j$ capture time and cell
fixed effects. By including fixed effects at the grid-cell level, we are able
to control for all time-invariant cross-section-specific factors impact-
ing migration such as the distance to ports or to the capital city,
distance to borders or the existing network in other regions or foreign
countries (i.e., presence of particular ethnic groups in destination
cities). In particular, the inclusion of such cell-specific intercepts
enables us to control for the existence of permanent differences in
birth and mortality rates at a very granular level of spatial disaggrega-
tion, thus allowing for an interpretation of our estimates as embodying
mostly effects on migration.

We also estimate a similar specification using land degradation
(proxied by soil carbon content) instead of drought, so as to empiri-
cally assess whether land degradation may have a direct effect on
migration. Contrary to our preferred indicator of drought (the SPEI
index), the interpretation of effects associated with this indicator is
potentially subject to reverse causality, as population density is likely
to impact land degradation and therefore its proxies. Further details
are provided in the Supplementary Information.

Our main explanatory variable of interest measures drought
events, potentially affecting livelihoods and the natural resources they
depend on. To capture such conditions, we make use of the SPEI[41]. The
SPEI is a multi-scalar drought index based on climatic data that are
normalized to mean zero and unit variance. It can be used for deter-
mining the onset, duration and magnitude of drought conditions with
respect to normal conditions in a variety of natural and managed
systems such as crops, ecosystems, rivers or water resources. A value
of zero implies that the water balance is exactly at its average; a value
of plus one (minus one) means that the water balance is one standard
deviation above (below) the average. The SPEI is constructed using an
array of weather, climate, and time-invariant factors that can measure
drought severity according to its intensity and duration, and can
identify the onset and end of drought episodes. Furthermore, the SPEI
allows comparison of drought severity through time and space, since it
can be calculated over a wide range of climatic zones.

For our grid-level analysis, we construct a measure of drought
periods as the proportion of months with SPEI scores below −1.5 out of
the past 12 months[42]. That is, for a year where the longest consecutive
streak of months below −1.5 is three, the cell will be given a value of
0.25. When the longest streak starts in the previous year, it is counted
and included in the year in which the streak ended. Theoretically, the
proportion can therefore be above unity. We aggregate this measure
for the past 2 years to capture longer drought periods, as grid-level
population data are only available every 5 years. Land degradation data
in the form of changes in soil carbon stocks comes from the Tren-
ds.earth project[43]. Specifically, we use soil carbon stock measures. In
order to estimate the potential tempering effects of alternative sour-
ces of income (mineral resources), we make use of the Mineral
Resources Data System (MRDS). The MRDS is sourced from a collec-
tion of reports describing metallic and nonmetallic mineral resources
throughout the world. The information provided includes deposit
name, location, commodity, deposit description, geologic character-
istics, production, reserves, resources, and references. We aggregate
deposit locations at the grid level to proxy resource and labor

availability. All the datasets are matched to the PRIO-GRID structure, a standardized spatial grid structure with global coverage at a resolution of decimal degrees[44]. The PRIO-GRID dataset is a grid structure that aids the compilation, management and analysis of spatial data within a time-consistent framework. It consists of quadratic grid cells that jointly cover all terrestrial areas of the world.

For each grid cell, we collect cell-specific information on armed conflicts, socioeconomic conditions, ethnic groups, geophysical attributes and climatic conditions. In our analysis, we use a number of those cell-specific attributes to investigate the potential heterogeneous effects of droughts and land degradation. These attributes include in particular information on natural resources (presence in a cell of oil, diamonds, gold or gems), distances between the centroid of the cell and international borders and to the capital city, as well as topographical features of the cell (whether it is mountainous terrain and its particular land composition). Our final dataset comprises information on 10,667 cells across 4 years, yielding a total of 42,936 observations. Over the full sample, the average cell has about 91,966 inhabitants. Furthermore, the average Gross Cell Product adjusted for purchasing power parities is about 0.1625 USD per cell with a maximum of 20.3 USD.

### Reporting summary
Further information on research design is available in the Nature Portfolio Reporting Summary linked to this article.

## Data availability
The original source of the international migration data used in the analysis is ref. [27]. The gridded population data are from the Gridded Population of the World (GPW) version 4 (https://sedac.ciesin.columbia.edu/data/collection/gpw-v4). The SPEI data are sourced from https://spei.csic.es/database.html and land degradation data in the form of changes in soil carbon stocks come from the Trends.earth project. Data on mineral resources are from the Mineral Resources Data System (MRDS, https://mrdata.usgs.gov/mrds/). All datasets are matched to the PRIO-GRID structure. All data used for the analysis can be found at the Harvard Dataverse under https://doi.org/10.7910/DVN/QXP0TY.

## Code availability
The replication code, written in Stata under version 16.1, is publicly available via Harvard Dataverse under https://doi.org/10.7910/DVN/QXP0TY.

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

## Acknowledgements

We would like to thank Michael Obersteiner and the participants at several workshops of the International Resource Panel for their helpful comments. J.C.C. acknowledges support from the eXplore! initiative, funded by the B&C Privatstiftung and Michael Tojner.

## Author contributions

M.B., J.C.C., D.K. and S.H.A. designed research. M.B. and J.C.C. analyzed data and performed research. M.B., J.C.C., D.K. and S.H.A. wrote the paper.

## Competing interests

The authors declare no competing interests.
