## [Peer Review File · Nature Communications]

Natural resources modulate the nexus between environmental shocks and human mobilityREVIEWER COMMENTS

Reviewer #1 (Remarks to the Author):

The paper identifies a critical knowledge gap around migration, immobility and slow-onset events such as drought and natural resource degradation and provides an innovative methodological approach. The paper adds to the existing literature and pushes the conversation on mobility choices forward. This is particularly helpful because of the vast disagreement in the literature on whether environmental shocks amplify or dampen migration. The focus on internal migration is particularly welcome. However, there are a few places where you can engage better with existing literature, temper some of the takeaways, and show more empirical detail to strengthen the paper.

I missed a conceptual figure tying together the causal links made (agricultural pathways to migration through natural resource degradation and drought). A figure tying these aspects together, with flows and scales shown (and where climate change, national mineral reserves etc. fit in) would be useful. Two papers that might help:

1. McLeman, R., Wrathall, D., Gilmore, E., Thornton, P., Adams, H., & Gemenne, F. (2021). Conceptual framing to link climate risk assessments and climate-migration scholarship. In *Climatic Change* (Vol. 165, Issues 1–2, pp. 1–7). *Climatic Change*. <https://doi.org/10.1007/s10584-021-03056-6>
2. de Haas, H. (2021). A theory of migration: the aspirations-capabilities framework. *Comparative Migration Studies*, 9(1), 8. <https://doi.org/10.1186/s40878-020-00210-4>
2. The argument of mineral resources and impacts on mobility makes sense but walking the reader through the linkages between the two would be helpful (with suitable references to back up this link).
3. Why were other resources such as national parks (with tourism benefits) for example not chosen?
4. How are multi-year droughts dealt with in the analysis? What about seasonal/cyclical movement? If there are data constraints around these, they need to be mentioned and some recognition of which types of mobility are covered (more long-term, permanent) and which are overlooked needs be made.
5. The broad continental findings are useful but would like to see some examples of intracontinental differences - perhaps maps of shifting population by cells.
6. The discussion section requires more engagement with the literature and policy arena. IOM is discussed but their mandate is not internal migration. International coordination is highlighted as a solution but perhaps local stewardship of natural resources might be more in tune with the findings on natural resource degradation. Lines 375-377 talk of drought duration but I don't think this is discussed in the findings. Overall, the discussion needs a rewrite.

Minor comments:

Check use of slightly different terms such as immobility in some places and non-migration in others - are you conveying the same idea through these or not?

Rewrite last line in the abstract which undersells the paper's findings and is also very difficult to follow!

Line 23 on pg 1 do you mean 'inability' instead of 'ability'?

Reviewer #2 (Remarks to the Author):

Thanks for the opportunity to review this paper. This paper builds on previous studies that tend to look at temperature and precipitation effects on international migration and immobility, and offers a more explicit and systematic (continent-scale) consideration of the effects of drought and landscape degradation on mobility. The study finds that drought causes internal migration but less so international, and recurrent drought can lead to more immobility, especially in poorer countries. I enjoyed reading the paper, and found it well written and concise in its presentation. But I had questions around the impacts of spatially variable uncertainty and accuracy of the datasets considered, and the choice of the degradation variable.

A more thorough discussion of the limitations and potential geographic biases in the population dataset is warranted. Some of the most immobile and climate-vulnerable populations include refugees, IDPs, and nomadic populations, but these are also some of the most likely to be excluded from census data collection that underpins GPW4 (see Carr-Hill 2013 <https://www.sciencedirect.com/science/article/pii/S0305750X13000053>). GPW4 has variable accuracy across countries with much of central and southern Africa having aggregated data from Level 2 census data compared to more disaggregated data in Kenya and S. Africa for example. The authors should account for this uncertainty in their model specifications. Finally, the 1/2 degree cell resolution is approximately 55km on each side at the equator. This aggregation overlooks the ability to assess the effect of drought on local (within-cell) mobility, which can include migration to urban areas from drought-affected rural agriculturally productive regions. The authors should clarify this limitation.

I suggest that the authors state that that the migrant stock dataset was used to assess the international migration in the main article. It's stated in the Supp Materials but was a lingering question when reading the main text -- how are they measuring international migration?

The timeframes of the datasets should be more directly stated. GPW is measured every five years since 2000, and international migration data are measured every 10 years from 1960 - 2000; these are clearly stated. But, over what time period(s) and frequency was change in SOC measured? What is the SPEI baseline for each effect estimate? It seems that the 20-year mean period used to construct the SPEI baseline might be different for international and internal migration? If so, the authors need to comment on any differences in SPEI trends between the different baseline periods since that would conceivably affect. The authors should clarify how these different datasets overlap in time.

In all models, it seems the authors included uninhabitable pixels such as lakes or glaciated regions from the analysis. It's not clear to me why these pixels were retained in the regressions.

For the land degradation assessment, the authors should add more information about how 'change in SOC' is actually measured. Assessing a single date SOC has non-negligible uncertainty which is spatially variable across the continent given sampling and modeling constraints. Measuring change in SOC is a function of discrete changes in land cover type such as converting grassland to built-up land (as described at https://trends.earth/docs/en/background/understanding_indicators15.html#soil-organic-carbon). The 'change in SOC' is really just a change in land cover then, which has its own endogenities with changes in population density and climatic effects. The authors need at least to explain what a measurement of 'change in SOC' entails. Further, the authors would do well to clarify whether a specific SOC depth was considered or an average, and show a map of SOC changes alongside Fig. 2 in Supplementary materials.

Given the uncertainties of measuring change in SOC, I was curious why the authors did not use another commonly used indicator of degradation: net primary productivity (NPP) (https://trends.earth/docs/en/background/understanding_indicators15.html#productivity). NPP is estimated annually across Africa with 500 meter resolution (e.g., <https://lpdaac.usgs.gov/products/mod17a3hv006/>) and should offer a more spatially and temporally consistent measure of degradation compared to SOC.

On line 40 of the main text, the authors state that they "identify immobile populations". Can the authors show where these immobile populations are located? Relatedly, what countries show high drought effect for international migration? Leaving out this information really limits the impact of the study's finding for targeted policy changes, as recommended in the final sentence of the Discussion.

There were a few minor typos in the paper as well:

- Line 206: I believe this should be topographic rather than topological
- "Droughth" in Table 3
- "soild" in Table 4 caption
- Table 4 caption mentions 5 columns but only includes 4
- In Supp Fig. 1, what is CSPEI used for x-axis? Country-level SPEI?

Reviewer #3 (Remarks to the Author):

Reviewer comments: "The Natural Resource Nexus with Human Migration Patterns and Immobile Populations"

The study combines high-resolution population density and climatic data to analyze the impact of drought events on migration in continental Africa. Further data on regional socioeconomic characteristics and the existence of mineral resources in an area is used to test for heterogeneity in the relationships. The authors show that while overall droughts increase migration, changes in population are more pronounced for more intense droughts and in poorer, less urban areas. Furthermore, adverse drought effects are dampened by the presence of mineral resources, possibly due to improved access of local populations to alternative income sources. Especially the latter result represents a novel finding as few studies have considered the climate-migration link also accounting for the presence of mineral resources. Overall, the article is well-written and employs rigorous empirical methods. It adds important and relevant insights to the field.

1. My main concern is related to the framing of the paper, which currently has a strong focus on immobility and trapped populations (e.g., in the abstract, introduction, and discussion). The authors write that no "systematic evidence of the phenomenon exists" and that they "assess quantitatively the prevalence of immobile populations". I do not see this as the main contribution of the paper, but rather its perspective on the natural resource nexus and the high level of detail and rigor of the analysis. As the authors rightly point out, studying immobile populations (in particular those that are trapped) is very challenging. Estimating non-monotonic effects of drought on migration is in my view not enough to comprehensively identify immobility or trapped populations, but can only serve as an indication. Even more, given that the analysis relies on population density as an imperfect measure of migration trends, the estimates might be affected by various other dynamics, which is also highlighted as a potential limitation by the authors. The increased effect sizes in poorer regions and for stronger drought events also rather speak against the notion of resource-constrained trapped populations. I would therefore recommend to put less emphasis on the immobility aspect (including in the title) and instead highlight the other contributions of the paper more strongly.

2. The authors present a number of heterogeneity analyses showing that the impact of drought on migration is moderated by drought intensity, income, urbanization, and the existence of mineral resources in an area. It would be interesting to estimate another suite of extended models simultaneously controlling for these different mediating channels. Especially, I would be interested to see a model that interacts at the same time the drought measure with the mineral resource measures and the income measure (drought + drought x minerals + drought x income). Lower migration responses in wealthier grid cells may not only be due to the existence of alternative sources of income, but also due to a generally higher resilience to shocks. By estimating a model with both interactions, these two channels could be tested for simultaneously.

3. In Table 4, the authors estimate models using country-year fixed effects resulting in a substantial reduction in the estimated drought effects. I do not think that these models are very

informative, given that the country-year fixed effects may capture most of the variation in the climatic variable, which is spatially highly correlated. I would either move these additional models as extended analyses to the supplement or remove them entirely from the paper. Running spatial models could represent an interesting extension of the paper adding also more insights on the spatial dependency in the relationships.

4. The findings on international migration could be featured more prominently in the main results section as they provide additional insights, especially on the detrimental effects of income as a moderator for internal versus international migration. Here, I was wondering why the authors decided to use two different approaches to measuring drought risks in an area (i.e., drought periods for internal, SPEI trajectories for international migration), which makes it difficult to directly compare the results. Also, the SPEI calculation for international migration considers only variations in SPEI during growing season. Would there be a possibility to more closely align the measurement approaches and analyses?

Minor points

5. I would be interested in the role of different types of mineral resources and their effects on the drought-migration links. Is there a difference between oil, mines, etc.? This could be considered in an extended analysis further breaking down the existence of mineral resources in the grid cell by different types.

6. "In line with (16), these results show statistically significant effects of drought on migration decisions (as captured in emigration rates), which are exclusively associated with relatively poor countries." (line 217) is hard to understand.

7. "Given the mean population by cell is about 91966, a reduction on 1% accounts for about 919 persons per cell, and a total reduction of 9.8 million people due to migration" (line 265). Does this reflect an estimate of the total size of drought-induced migration across all grids? If so, then this could be highlighted more strongly.

8. Parts from the main text (e.g., on measurement details, 150ff) could be moved to the methods section. Part of the supplementary materials (e.g. line 38ff) are redundantly repeating the main text.

RESPONSE TO COMMENTS BY REVIEWER #1

Let us start by thanking you for your thoughtful comments, which have helped us improve the manuscript significantly. Below you find the answer to each one of them, including a description of how we modified the paper to account for them.

I missed a conceptual figure tying together the causal links made (agricultural pathways to migration through natural resource degradation and drought). A figure tying these aspects together, with flows and scales shown (and where climate change, national mineral reserves etc. fit in) would be useful. Two papers that might help:

1. McLeman, R., Wrathall, D., Gilmore, E., Thornton, P., Adams, H., & Gemenne, F. (2021). Conceptual framing to link climate risk assessments and climate-migration scholarship. In *Climatic Change* (Vol. 165, Issues 1–2, pp. 1–7). *Climatic Change*. <https://doi.org/10.1007/s10584-021-03056-6>

2. de Haas, H. (2021). A theory of migration: the aspirations-capabilities framework. *Comparative Migration Studies*, 9(1), 8. <https://doi.org/10.1186/s40878-020-00210-4>

We have now included a conceptual figure (see below) summarizing the theoretical framework and the causal linkages implied by the interaction between resource scarcity and migration. This is the new Figure 1 in the revised manuscript and should help the reader understand the theoretical context in which our empirical analysis is carried out. The figure emphasizes how resource scarcity and abundance intersect with migration patterns, with minerals constituting an example of resource abundance which can lead to "rushes" and thus concomitantly pose governance challenges.

2. The argument of mineral resources and impacts on mobility makes sense but walking the reader through the linkages between the two would be helpful (with suitable references to back up this link).

We have expanded the description of the mobility impacts of (mineral) resources in the revised text, linking the discussion to further references and to the conceptual framework summarized in the new Figure 1.

3. Why were other resources such as national parks (with tourism benefits) for example not chosen?

The link we investigate in our analysis acts through an agricultural and mining pathway and highlights the role played by drought on land degradation and thus on income potential for households relying on agricultural returns. In this respect, national parks do not fit into the narrative employed in our theoretical framework, although we agree that they would be an interesting object of study in further research efforts, and are now mentioned in the text of the revised manuscript as a potentially interesting object of future research.

4. How are multi-year droughts dealt with in the analysis? What about seasonal/cyclical movement? If there are data constraints around these, they need to be mentioned and some recognition of which types of mobility are covered (more long-term, permanent) and which are overlooked needs be made.

Multi-year seasonal/cyclical droughts are captured in our methodology as (recurring) short-duration episodes (see the method for identifying drought duration in the Supplementary Material). In the original manuscript we did not refer explicitly to such phenomena, we have now included a reference to them when presenting the results of the analysis. Unfortunately, the data on migration does not allow us to differentiate between long-term, permanent mobility and return migration patterns. We explicitly mention this limitation of the analysis in the revised manuscript and propose the use of more detailed data for case studies that may shed light on the heterogeneous effect of environmental shocks on these different types of mobility phenomena (see revised Discussion section).

5. The broad continental findings are useful but would like to see some examples of intracontinental differences - perhaps maps of shifting population by cells.

We now include a map of population growth by grid cell over the full period in the revised manuscript.

6. The discussion section requires more engagement with the literature and policy arena. IOM is discussed but their mandate is not internal migration. International coordination is highlighted as a solution but perhaps local stewardship of natural resources might be more in tune with the findings on natural resource degradation. Lines 375-377 talk of drought duration but I don't think this is discussed in the findings. Overall, the discussion needs a rewrite.

We have extensively rewritten the discussion section, expanding it to also cover issues related to ownership of natural resources and linking it more directly to the literature on global governance challenges, including studies on stewardship of natural resources.

Minor comments:

Check use of slightly different terms such as immobility in some places and non-migration in others - are you conveying the same idea through these or not?

Rewrite last line in the abstract which undersells the paper's findings and is also very difficult to follow!

Line 23 on pg 1 do you mean 'inability' instead of 'ability'?

We have amended all of these issues in the revised manuscript. Thanks again for your careful reading of our paper and the extremely useful comments.

RESPONSE TO COMMENTS BY REVIEWER #2

Let us start by thanking you for your thoughtful comments, which have helped us improve the manuscript significantly. Below you find the answer to each one of them, including a description of how we modified the paper to account for them.

A more thorough discussion of the limitations and potential geographic biases in the population dataset is warranted. Some of the most immobile and climate-vulnerable populations include refugees, IDPs, and nomadic populations, but these are also some of the most likely to be excluded from census data collection that underpins GPW4 (see Carr-Hill 2013 <https://www.sciencedirect.com/science/article/pii/S0305750X13000053>). GPW4 has variable accuracy across countries with much of central and southern Africa having aggregated data from Level 2 census data compared to more disaggregated data in Kenya and S. Africa for example. The authors should account for this uncertainty in their model specifications. Finally, the 1/2 degree cell resolution is approximately 55km on each side at the equator. This aggregation overlooks the ability to assess the effect of drought on local (within-cell) mobility, which can include migration to urban areas from drought-affected rural agriculturally productive regions. The authors should clarify this limitation.

We have now included all of these limitations of the dataset employed more explicitly in the text of the revised manuscript, as well as a reference to Carr-Hill (2013).

I suggest that the authors state that that the migrant stock dataset was used to assess the international migration in the main article. It's stated in the Supp Materials but was a lingering question when reading the main text -- how are they measuring international migration?

We passed this information to the main text in the revised manuscript, where we now provide more information about the analysis of the effect of droughts on international migration flows.

The timeframes of the datasets should be more directly stated. GPW is measured every five years since 2000, and international migration data are measured every 10 years from 1960 - 2000; these are clearly stated. But, over what time period(s) and frequency was change in SOC measured? What is the SPEI baseline for each effect estimate? It seems that the 20-year mean period used to construct the SPEI baseline might be different for international and internal migration? If so, the authors need to comment on any differences in SPEI trends between the different baseline periods since that would conceivably affect. The authors should clarify how these different datasets overlap in time.

We are explicit in the description of the time dimension of the analysis in the revised manuscript. Our international migration analysis spans the period 1960-2000 in 10-year intervals, while the grid-level analysis spans the period 2000-2015 in 5-year intervals.

In all models, it seems the authors included uninhabitable pixels such as lakes or

glaciated regions from the analysis. It's not clear to me why these pixels were retained in the regressions.

We use GPW data. The Gridded Population of the World (GPW) collection, now in its fourth version (GPWv4), models the distribution of human population (counts and densities) on a continuous global raster surface. We match this data with climate data of similar resolution. Using log transformed population projections, we effectively drop locations that are not inhabited - or projected to be inhabited.

For the land degradation assessment, the authors should add more information about how 'change in SOC' is actually measured. Assessing a single date SOC has non-negligible uncertainty which is spatially variable across the continent given sampling and modeling constraints. Measuring change in SOC is a function of discrete changes in land cover type such as converting grassland to built-up land (as described at https://trends.earth/docs/en/background/understanding_indicators15.html#soil-organic-carbon). The 'change in SOC' is really just a change in land cover then, which has its own endogenities with changes in population density and climatic effects. The authors need at least to explain what a measurement of 'change in SOC' entails. Further, the authors would do well to clarify whether a specific SOC depth was considered or an average, and show a map of SOC changes alongside Fig. 2 in Supplementary materials.

Given the uncertainties of measuring change in SOC, I was curious why the authors did not use another commonly used indicator of degradation: net primary productivity (NPP) (https://trends.earth/docs/en/background/understanding_indicators15.html#productivity). NPP is estimated annually across Africa with 500 meter resolution (e.g., <https://lpdaac.usgs.gov/products/mod17a3hv006/>) and should offer a more spatially and temporally consistent measure of degradation compared to SOC.

For this study, we chose to retain only the last sub-indicator, soil carbon stock degradation, because land productivity and land use change are very likely to be directly and mechanically impacted by population density, which is our output variable in the regression model. Changes in carbon stock are slow moving and, by construction, less likely to suffer directly from this reverse causality bias. Carbon stock is the quantity of carbon in a reservoir which has the capacity to accumulate or release carbon and comprised of above and below-ground biomass, dead organic matter, and soil organic carbon. The original source is the Soilgrids.org database (described in Hengl et al., 2017). It provide soil organic carbon content for the entire globe at 250 metres resolution and for 7 different soil depth (0 cm, 5 cm, 15 cm, 30 cm, 60 cm, 100 cm and 200 cm).

The trend.earth project approximates soil carbon stock degradation between periods by multiplying changes in soil organic carbon by a set of conversion coefficients. Areas experiencing a decrease of at least 10% of their soil organic carbon stocks are considered degraded. Our degradation variable simply takes the share degraded within each cell. We have added this information to the Supplementary Material in the revised manuscript.

We have considered net primary productivity changes in alternative specifications. These data have very minimal variation across time in the available source (and a limited temporal coverage) which prevents us from obtaining meaningful results based on within-cell variability.

On line 40 of the main text, the authors state that they "identify immobile populations". Can the authors show where these immobile populations are located? Relatedly, what

countries show high drought effect for international migration? Leaving out this information really limits the impact of the study's finding for targeted policy changes, as recommended in the final sentence of the Discussion.

In the revised manuscript we include more information in this direction. On the one hand, we present a population growth map at the grid-cell level, which allows the reader to grasp population dynamics at a granular level in the African continent. On the other hand, we now provide a heat map showing the differences in strength of the effect of drought shocks on mobility we find in our regressions. We describe these results in the revised text and link them to the discussion on immobility. In particular, the figure depicting the effect size implied by the interaction terms should shed light on the identification of restrictions to human mobility following environmental shocks.

There were a few minor typos in the paper as well:

- **Line 206: I believe this should be topographic rather than topological**
- **"Droughth" in Table 3**
- **"soild" in Table 4 caption**
- **Table 4 caption mentions 5 columns but only includes 4**
- **In Supp Fig. 1, what is CSPEI used for x-axis? Country-level SPEI?**

We corrected the typos and clarified the issues mentioned. CSPEI refers to growing-season SPEI, we describe the acronym in the text of the figure.

Thanks again for your careful reading of our paper and for your insightful comments, that have helped us significantly improve the paper.

RESPONSE TO COMMENTS BY REVIEWER #3

Let us start by thanking you for your thoughtful comments, which have helped us improve the manuscript significantly. Below you find the answer to each one of them, including a description of how we modified the paper to account for them.

1. My main concern is related to the framing of the paper, which currently has a strong focus on immobility and trapped populations (e.g., in the abstract, introduction, and discussion). The authors write that no “systematic evidence of the phenomenon exists” and that they “assess quantitatively the prevalence of immobile populations”. I do not see this as the main contribution of the paper, but rather its perspective on the natural resource nexus and the high level of detail and rigor of the analysis. As the authors rightly point out, studying immobile populations (in particular those that are trapped) is very challenging. Estimating non-monotonic effects of drought on migration is in my view not enough to comprehensively identify immobility or trapped populations, but can only serve as an indication. Even more, given that the analysis relies on population density as an imperfect measure of migration trends, the estimates might be affected by various other dynamics, which is also highlighted as a potential limitation by the authors. The increased effect sizes in poorer regions and for stronger drought events also rather speak against the notion of resource-constrained trapped populations. I would therefore recommend to put less emphasis on the immobility aspect (including in the title) and instead highlight the other contributions of the paper more strongly.

Following your advice, in the revised manuscript we have changed the title and tilted the focus of the narrative from immobile populations to the role of resources as a mediator between climatic shocks and human mobility. We feel that framing the analysis within the discussion of immobility is important and that our paper can contribute to this branch of the literature by adding nuance to mineral-rush migration and assessing how internal movements within national boundaries could lead to a reduction in international migration under specific conditions. In this sense, we have kept some of the discussion on immobile populations in the text, but now as one issue within the broader context of the resource nexus. We also redrafted the discussion section in the same lines. In addition, in response to the concerns of Reviewer #1, we also present a more cohesive theoretical framework to embed our empirical analysis in the introductory section (see Figure 1).

2. The authors present a number of heterogeneity analyses showing that the impact of drought on migration is moderated by drought intensity, income, urbanization, and the existence of mineral resources in an area. It would be interesting to estimate another suite of extended models simultaneously controlling for these different mediating channels. Especially, I would be interested to see a model that interacts at the same time the drought measure with the mineral resource measures and the income measure (drought + drought x minerals + drought x income). Lower migration responses in wealthier grid cells may not only be due to the existence of alternative sources of income, but also due to a generally higher resilience to shocks. By estimating a model with both interactions, these two channels could be tested for simultaneously.

We have now estimated the specification you propose, and included it in the text. The results indicate that the difference in the strength of the effect is better explained by differences in income level, and that once we control for economic development, the interaction term with

the resource covariate appear significant. The correlation between income and mineral resources is likely to be driving the insignificant effects found for the resource when we allow for different sources of parameter heterogeneity in the regression specification. The results of this regression model are included in the Supplementary Material of the revised version of the paper and briefly discussed in the text.

3. In Table 4, the authors estimate models using country-year fixed effects resulting in a substantial reduction in the estimated drought effects. I do not think that these models are very informative, given that the country-year fixed effects may capture most of the variation in the climatic variable, which is spatially highly correlated. I would either move these additional models as extended analyses to the supplement or remove them entirely from the paper. Running spatial models could represent an interesting extension of the paper adding also more insights on the spatial dependency in the relationships.

We included these particular regression specifications in order to show the robustness of the overall results found and due to the comment of a previous reviewer. The casual reader may argue that the results from models without fixed effects at the country and year level could be driven by (unobservable) differences across countries or by global shocks to the continent. For the sake of completeness, and in order to avoid this criticism, we would propose to leave the models in the text. If you insist on taking these results from the main text, we would of course be willing to do it. In the models estimated, the spatial component is assumed to be subsumed in the error term and controlled for using the spatial HAC correction on the standard error of the estimates. The estimation of explicit spatial models (with a spatial autoregressive structure, for instance) would be an interesting and potentially fruitful avenue of further research, which we put forward in the discussion section. The additional methodological complexities involved in the estimation of such models (for instance, specification choice in the presence of uncertainty about the nature of the –possibly time-varying – spillovers) imply that its inclusion in the current manuscript would go beyond the scope of our contribution.

4. The findings on international migration could be featured more prominently in the main results section as they provide additional insights, especially on the detrimental effects of income as a moderator for internal versus international migration. Here, I was wondering why the authors decided to use two different approaches to measuring drought risks in an area (i.e., drought periods for internal, SPEI trajectories for international migration), which makes it difficult to directly compare the results. Also, the SPEI calculation for international migration considers only variations in SPEI during growing season. Would there be a possibility to more closely align the measurement approaches and analyses?

We have expanded the presentation of the results on international migration in the main text of the manuscript.

Minor points

5. I would be interested in the role of different types of mineral resources and their effects on the drought-migration links. Is there a difference between oil, mines, etc.?

This could be considered in an extended analysis further breaking down the existence of mineral resources in the grid cell by different types.

We have included this point as a potentially interesting expansion of the analysis in the discussion, although data limitations might be an obstacle to carrying out such an exercise for very fine disaggregations of mineral resources.

6. “In line with (16), these results show statistically significant effects of drought on migration decisions (as captured in emigration rates), which are exclusively associated with relatively poor countries.” (line 217) is hard to understand.

We have rephrased this sentence.

7. “Given the mean population by cell is about 91966, a reduction on 1% accounts for about 919 persons per cell, and a total reduction of 9.8 million people due to migration” (line 265). Does this reflect an estimate of the total size of drought-induced migration across all grids? If so, then this could be highlighted more strongly.

We now mention that this is the size implied by our estimates across all grids.

8. Parts from the main text (e.g., on measurement details, 150ff) could be moved to the methods section. Part of the supplementary materials (e.g. line 38ff) are redundantly repeating the main text.

We have moved the measurement details to the methods section and shortened the supplementary materials to avoid repetition.

REVIEWERS' COMMENTS

Reviewer #2 (Remarks to the Author):

I congratulate the authors on a clearly written and helpful contribution. I appreciate the reduced emphasis and additional clarity on immobile populations in the revised manuscript. With the additional revisions suggested by the other reviewers, this version has satisfactorily addressed my concerns. My only two remaining comments concern the figures. Fig. 1 needs to be sharper, and is mis-cited as Fig. 4 in the text. Fig. 4 needs a N/A data value in the legend, and text explaining the reason for the N/A.

Reviewer #3 (Remarks to the Author):

The authors have comprehensively responded to all my previous comments and have further developed and improved the manuscript. I have a few minor comments left, which should still be addressed before the publication of the article.

The authors included a conceptual figure which I found useful but at the same time a bit confusing. It was not clear to me why resource abundance influences the wealth distribution and economic development, but not the other way around. Also, how political governance and security management challenges come into play here was not clear from the depiction. Generally, the used terminology in the figure is a bit technical (e.g. resource scramble, rentier state, carrying capacity) and it would be good to explain and elaborate on some of the used terms and concepts in the text or in the figure caption.

When reading the manuscript again, I felt that parts of the text, in particular in the introduction (e.g. paragraph line 60ff & paragraph line 111ff) could benefit from some simplification, e.g. by using less technical terminology and putting a stronger emphasis on key aspects of the study.

The y-axis label is missing in Fig3. I assume the axes show the marginal effects of the models, but this is not clear from the depiction. The authors could generally expand the figure captions to provide more relevant information about the figures.

Reviewer #2: I congratulate the authors on a clearly written and helpful contribution. I appreciate the reduced emphasis and additional clarity on immobile populations in the revised manuscript. With the additional revisions suggested by the other reviewers, this version has satisfactorily addressed my concerns. My only two remaining comments concern the figures. Fig. 1 needs to be sharper, and is mis-cited as Fig. 4 in the text. Fig. 4 needs a N/A data value in the legend, and text explaining the reason for the N/A.	Thanks a lot, and thanks again for your helpful comments. We have improved the resolution of Figure 1, corrected the typo in the reference to this figure in the text and explained the missing values of Figure 4 in the corresponding caption.
Reviewer #3 (Remarks to the Author): The authors have comprehensively responded to all my previous comments and have further developed and improved the manuscript. I have a few minor comments left, which should still be addressed before the publication of the article. The authors included a conceptual figure which I found useful but at the same time a bit confusing. It was not clear to me why resource abundance influences the wealth distribution and economic development, but not the other way around. Also, how political governance and security management challenges come into play here was not clear from the depiction. Generally, the used terminology in the figure is a bit technical (e.g. resource scramble, rentier state, carrying capacity) and it would be good to explain and elaborate on some of the used terms and concepts in the text or in the figure caption. When reading the manuscript again, I felt that parts of the text, in particular in the introduction (e.g. paragraph line 60ff & paragraph line 111ff) could benefit from some simplification, e.g. by using less technical terminology and putting a stronger emphasis on key aspects of the study. The y-axis label is missing in Fig3. I assume the axes show the marginal effects of the models, but this is not clear from the depiction. The authors could generally expand the figure captions to provide more relevant information	Thanks you very much, and thanks again for your helpful comments in previous versions of the manuscript. We have now revised and slightly extended the description of the conceptual figure to make it more accessible to a broader readership. We simplified the text in these parts of the introduction and streamlined the explanation of the contribution of the paper. The revised introduction should be significantly more accessible in its current revised version. We included the labeling of the axis and expanded the figure caption for this particular graph. We also included more information in the captions of the other figures of the text.

about the figures.	
--